# A risk scoring model of COVID-19 at hospital admission

**João José Ferreira Gomes** [1] *, **António Ferreira**[2], **Afonso Alves** [1], **Beatriz Nogueira Sequeira**[1]

1 Faculdade de Ciências, Universidade de Lisboa, Lisboa, Portugal, 2 Universidade de Trás-os-Montes e Alto Douro, Vila Real, Portugal

* jjgomes@campus.ul.pt

## Abstract

### Background

The COVID-19 pandemic has been the most serious public health crisis in recent times, a pandemic whose impact was felt across the globe in various groups and populations. Confronted with an urgent problem, people and governments were forced to make decisions without fully understanding the disease. The present work aims to reinforce our ever-growing knowledge of the illness, particularly in modelling the risk of death of a patient admitted to a hospital with a positive COVID-19 test.

### Methods

Given the simplicity of using and programming logistic regression in any national healthcare unit and the ease of interpreting the results, we chose to use this technique over several other. Using scoring techniques, it is possible to associate the various diagnoses with a numerical value (score), making it possible therefore to integrate the patient's multiple medical conditions as a single continuous variable in the model.

### Results

It is possible to establish with good discriminatory capacity (ROC AUC Test = 0.8) which COVID patients are at higher risk when admitted to the healthcare unit—people of advanced age with pre-existing conditions, such as diabetes and high blood pressure, or newly acquired conditions, such as pneumonia. Moreover, males and clinical episodes occurring in healthcare units with few available beds (high healthcare unit occupancy) are also at higher risk. The importance of each variable in predicting the target is: age (47%), sum of comorbidity scores (28%), healthcare unit score (12.0%), gender score (7%) and healthcare unit occupancy (6%).

### Conclusions

Using a dataset with more than 52000 people, it was possible to successfully differentiate likelihood of death by COVID using age, comorbidity information, healthcare unit, healthcare unit occupancy and gender. The age and the comorbidities associated with each patient

**Data Availability Statement:** We made the data and R scripts available at this github link: https://github.com/jjfgomes/A-risk-scoring-model.

**Funding:** The first author was supported by FCT – Fundação para a Ciência e a Tecnologia, under the project: UIDB/04561/2020.

**Competing interests:** The authors have declared that no competing interests exist.

had a joint contribution of about 75% in explaining the COVID related mortality in Portuguese public hospitals in the period between March 2020 and May 2021.

## Introduction

The Coronavirus disease (COVID-19) caused by severe acute respiratory syndrome (SARS-COV-2) raised the biggest global alarm since the Spanish Flu pandemic of 1918. The images that first came from China, followed by those in the Bergamo area in Italy, instantly led to a generalized state of fear among people resulting in the implementation of lockdowns by many governments (especially in the developed countries), affecting millions of people [1].

The consequences were serious both mentally and economically [2, 3], and the measures taken did not always seem to have the intended effect [4]. Panic and a growing number of cases led to difficulties in accessing reliable information crucial to better understand the dynamics of the disease and the clinical determinants of its gravity.

It is challenging to determine with the existing data the precise risk level of a patient admitted to the hospital with SARSCOV-2. It is important to understand how serious was the risk of death from SARSCOV-2 for people who came to health services every day with symptoms of this disease. There are several articles on this topic [5, 6].

The database we had access to comes from the Shared Services of Ministry of Health (SPMS, Portugal) and is characterized by episodes (an episode corresponds to a hospital admission and the same person can be associated with more than one episode). The patients' privacy is assured through reference codes that we transformed into ID's (1, 2, 3, . . .). Each episode is described with a referral healthcare unit (cluster of hospitals), date of birth, gender, start date, end date (which can be categorized in death in the hospital or recovery) and the various clinical diagnoses in addition to SARSCOV-2 associated with that episode (comorbidities). Each clinical diagnosis corresponds to a row in the database.

## Methods

Thanks to SPMS we were able to access data that until May 2021 are the most reliable and complete in Portugal. The data provided by SPMS include patients who between 1 March 2020 and 10 May 2021 were admitted to a Portuguese public hospital and, at some point during that period, had a positive diagnosis of COVID.

This paper will present a model that allows us to evaluate the probability of survival for a patient who was hospitalized in Portugal, based on variables including age, healthcare unit, average healthcare unit occupancy over the patient's stay period, gender, and other clinical diagnoses. This information could enable doctors to decide, for instance, whether a patient should enter the ICU (intensive care unit).

Some small healthcare units and patients who periodically went to the hospital for haemodialysis were not considered. In addition, all non-adult patients were excluded since it was found that SARSCOV-2 does not damage the immune system of this type of patient except in very specific cases without a defined pattern [7]. (Fig 1).

Regarding the clinical diagnoses we were able, with the help of a medical practitioner, to exclude comorbidities that were not significant to our study and consider only those that would contribute to a worse diagnosis when associated with a respiratory disease.

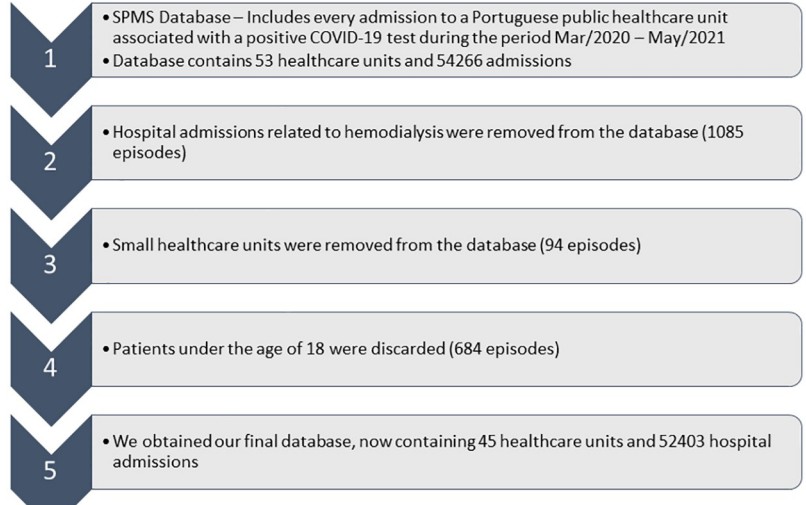

**Fig 1. Presentation of the filters applied to the initial database.**

We were unable to rule out patients who tested positive for COVID in the hospital but were admitted for other causes, as the data do not contain this information. Furthermore, we were unable to filter out cases where a patient might have COVID but died due to other causes. Therefore, this study focuses on patients admitted to public hospitals who tested positive for COVID at some point in the procedure.

There follows a summary of the variables available to help us understand the target variable, namely "Death".

## Data

The variables available were: Code identification by episode (id), Healthcare unit (anonymized) (hcu), Capacity (number of beds in the healthcare unit) (hcu_dim), Date of admission (in_date), Age by year (age), Gender, Outcome after discharge (Survival (0) or Death (1)), Clinical Diagnoses with description, Length of Stay by episode in days (los).

We refer to the target variable as "death"–with 52,403episodes of which 12,546 (24%) resulted in the patient's death.

Initially we have four predictor variables–age, gender, clinical diagnoses and hcu.

Moreover, considering the entry and exit dates of each patient and healthcare unit capacity, it was possible to assess the daily healthcare unit prevalence and, from there, to understand the level of healthcare unit occupancy (average healthcare unit occupancy rate) to which each patient was subject to during their hospitalization period: number of patients in the database/ number of beds (information in National Health Service [8]). The hcu occupancy will be our $5^{th}$ predictor variable.

Let's now consider the two hcu-related variables, represented in Figs 2 and 3.

We note that, although we know the number of beds available for each health unit, it is unknown how many of these beds can actually be occupied by COVID patients.

Besides the two healthcare unit-related variables presented, we have three patient-related variables–age (continuous), clinical diagnoses and gender (categorical). We will be describing these variables in the Results section.

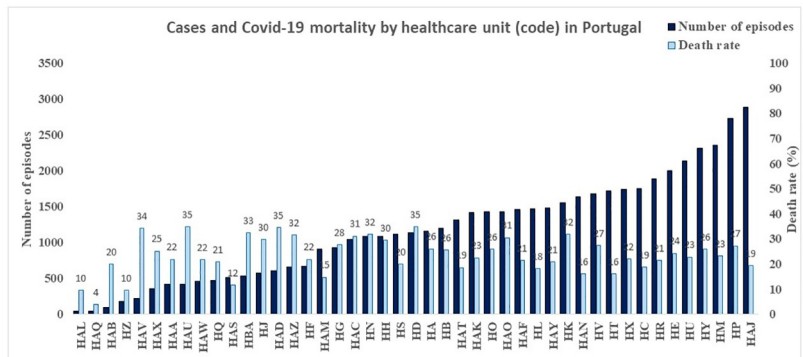

**Fig 2. Cases and Covid-19 mortality by healthcare units (code) in Portugal.**

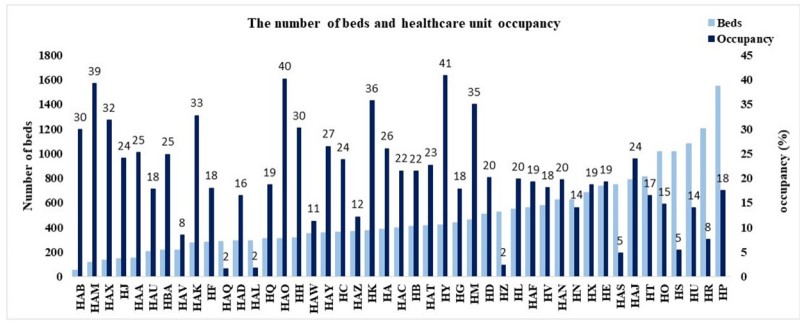

**Fig 3. Healthcare unit occupancy and number of beds available for each of the healthcare units in our database.**

## Statistical modelling

In our preliminary steps, we will be transforming the categorical variables into scores, a technique that is widely used for the probability of default in bank loans (see for example [9–11]). Thus, we will work with 5 continuous variables.

This method follows a scorecard model approach, which is meant to be simpler to explain and apply. Two of the most common cited references in the literature are for instance [10, 11]. Let's take variable hcu, for example, that contains 45 categories. Considering that one of the dummy variables always serves as a reference category we need to estimate 44 parameters in total (excluding the intercept variable, other categorical variables and potentially other numerical variables). In our case, we are dealing with 3 categorical variables: hcu (with 45 categories), clinical diagnoses (27 comorbidities) and gender (2 categories), besides the 2 continuous variables age and hcu occupancy. For the categorical variables one can apply a monotonic transformation related to the target variable to obtain a new numerical variable with a single corresponding coefficient. This transformation can either be monotonically increasing or decreasing resulting in either a positive or negative value of the beta parameter, respectively. The transformation of the categorical variable into numerical with a single coefficient results in a more parsimonious model, improving interpretability.

In the special case of the comorbidity variable (clinical diagnoses), we started by calculating a score for each diagnosis. Following the score estimation, the scores were adjusted to the range [0;1] so that they no longer contained negative values (typical score transformation generates positive and negative scores). Thus, having more comorbidities can only increase the

risk of death. Finally, we added the scores by episode to obtain episode specific information (for example, for a patient/episode with 5 comorbidities we would sum the corresponding 5 scores). Thus, the score transformation in the case of the comorbidities enables us to account for multiple comorbidity patient profiles, a feature not easily achievable using just the original categorical information. Although this is an innovative approach, we highlight two potential problems with this variable: (i) the different comorbidities are combined additively per episode, but this is not the only approach (e.g. the multiplicative model would be a valid alternative); (ii) the score per comorbidity is created univariately, which is an expedient approach, but a patient with two comorbidities of low severity, when combined, may have high severity (and by adding two low scores we always get a low total score).

We split the dataset into training (70%) and testing (30%) sets. The scores for each healthcare unit, gender and comorbidity were computed in the training to avoid information leakage from the test partition.

Having 5 continuous variables, standardization was applied: each variable was transformed to have null mean and unit variance (the standardization parameters, the mean and variance, were computed in the training partition). The standardization makes it possible to compare the model coefficients and infer the relative importance of the different variables.

## Results

After being filtered, the database represents 45 Portuguese public healthcare units containing 52,403 episodes. Besides the variables related to hcu, we are also considering variables related to the patients–age, gender and clinical diagnoses–as we can observe in Table 1.

After exploring the variables, we proceed to the construction of our model, starting with univariate regression.

### Univariate regression

Logistic regression allows us, unlike new data science tools, to obtain a prediction and respective confidence interval for new patients. In Table 2 we present, still in a univariate way, the variables available to estimate the probability of death at hcu admission:

In the following section we will go through the process of building the Logistic Regression model.

### Multiple regression

We will start by the 4 variables that contain a single piece of information per episode, and later introduce the 5th variable, clinical diagnoses (Table 3), of which the same episode can have several associated values–one for each comorbidity.

Since all variables have the same scale (due to standardization), we can compare and rank the contribution of each variable through its regression coefficient [12] (a variable with higher absolute coefficient value has higher model impact). If we take one step further and assume that the sum of the absolute coefficients represents the total modelled effects, we can relativize the importance of each variable $j$ by dividing each coefficient by this sum (Table 4). This practice is common in a scorecard model approach [10, 11]:

Subsequently, we tried to understand if there were any relevant interactions between the variables, and we found two that revealed to be somewhat important, as they led to a slight predictive improvement.

These interactions express that gender and comorbidities may have a different impact depending on the patient's age. Thus, we obtained the model in Table 5:

**Table 1. Descriptive statistics for four of the variables used in this study.**

| Variables | Categories | Total | Death | Rate |
|---|---|---|---|---|
| Age (mean = 70.7, sd = 17.1, median = 74.0) | [18_40] | 3535 | 51 | 0.01 |
| | [40_50] | 3617 | 161 | 0.04 |
| | [50_60] | 6033 | 457 | 0.08 |
| | [60_70] | 9323 | 1372 | 0.15 |
| | [70_80] | 11628 | 3025 | 0.26 |
| | [80_85] | 7296 | 2540 | 0.35 |
| | [85_90] | 6535 | 2724 | 0.42 |
| | [90_95] | 3415 | 1656 | 0.48 |
| | [95_100] | 940 | 517 | 0.55 |
| | [100_105] | 81 | 43 | 0.53 |
| Gender | Male | 24777 | 5651 | 0.23 |
| | Fem | 27626 | 6895 | 0.25 |
| Clinical Diagnoses | Smoking (TABACO) | 2261 | 443 | 0.20 |
| | Obesity (OB) | 9234 | 1832 | 0.20 |
| | Arterial hypertension (HTA) | 22119 | 5004 | 0.23 |
| | COVID-19 (COVID) | 54384 | 12607 | 0.23 |
| | Viral pneumonia (PV) | 12589 | 2946 | 0.23 |
| | Pulmonary thromboembolism (TEP) | 1565 | 399 | 0.25 |
| | Hyponatremia (HIPONA) | 3100 | 821 | 0.26 |
| | Diabetes mellitus (DM) | 19787 | 5280 | 0.27 |
| | Pneumonia to SARSCoV (PCOV) | 22409 | 6399 | 0.29 |
| | Non-pulmonary localized infection (ILNP) | 6619 | 1928 | 0.29 |
| | Anemia (ANE) | 7096 | 2174 | 0.31 |
| | Acute abdominal disease (DAA) | 2074 | 642 | 0.31 |
| | Chronic respiratory disease (DPCO) | 6285 | 2012 | 0.32 |
| | Acute cerebrovascular disease (AVC) | 5368 | 1798 | 0.33 |
| | Ischemic heart disease (EM) | 2650 | 929 | 0.35 |
| | Chronic kidney failure (IRC) | 14506 | 5156 | 0.36 |
| | Bacterial pneumonia (PB) | 10751 | 3887 | 0.36 |
| | Acute breathing insufficiency (IRESPA) | 14800 | 5531 | 0.37 |
| | Pulmonary hypertension (HTP) | 715 | 269 | 0.38 |
| | Cardiac insufficiency (IC) | 13017 | 4936 | 0.38 |
| | Atrial fibrillation (FA) | 7901 | 3015 | 0.38 |
| | Acute kidney failure (IRA) | 8974 | 3784 | 0.42 |
| | Coagulation changes (AC) | 377 | 165 | 0.44 |
| | Neoplastic disease (cancer) (NEO) | 6846 | 3070 | 0.45 |
| | Liver failure (IH) | 840 | 377 | 0.45 |
| | Fungal pneumonia (PF) | 90 | 49 | 0.54 |
| | Septicemia (SEPSIS) | 3517 | 2255 | 0.64 |
| Occupancy (%) (mean = 22.7, sd = 15.1, median = 20.7.0) | [0_10] | 12232 | 2294 | 0.19 |
| | [10_20] | 13768 | 3242 | 0.24 |
| | [20_30] | 13887 | 3615 | 0.26 |
| | [30_80] | 12516 | 3395 | 0.27 |

**Table 2. Results of fitting univariate logistic regression model.**

| Variable | Estimated Coeff | Standard Error | z | p-value | OR (95% CI) |
|---|---|---|---|---|---|
| age | 1.143 | 0.018 | 62.52 | <0.001 | 3.13 (3.02–3.25) |
| gender | 0.066 | 0.012 | 5.54 | <0.001 | 1.07 (1.04–1.09) |
| hcu_occupancy | 0.168 | 0.011 | 14.71 | <0.001 | 1.37 (1.33–1.4) |
| healthcare unit | 0.311 | 0.012 | 25.34 | <0.001 | 1.18 (1.16–1.21) |
| Clinical diagnoses | 0.783 | 0.013 | 62.12 | <0.001 | 2.19 (2.13–2.24) |

All variables have a very low p-value, proving to be potentially useful in predicting death in a logistic regression model.

**Table 3. Model built from the principal variables and with the training dataset.**

| Variable | Estimated Coeff | Standard Error | z | p-value | OR (95% CI) |
|---|---|---|---|---|---|
| (Intercept) | -1.64117 | 0.01752 | -93.68 | <2e-16 | |
| age | 1.106 | 0.021 | 53.63 | <2e-16 | 3.02 (2.9–3.15) |
| gender | 0.175 | 0.014 | 12.67 | <2e-16 | 1.19 (1.16–1.22) |
| hcu_occupancy | 0.151 | 0.013 | 11.38 | <2e-16 | 1.16 (1.13–1.19) |
| healthcare unit | 0.280 | 0.014 | 20.02 | <2e-16 | 1.32 (1.29–1.36) |
| Clinical diagnoses | 0.664 | 0.013 | 49.81 | <2e-16 | 1.94 (1.89–1.99) |

**Table 4. Relative importance of each variable for the model.**

| $coef_j/\Sigma_i|coef_i|$ | | | | |
|---|---|---|---|---|
| age | gender | hcu_occupancy | healthcare unit | Clinical diagnoses |
| 46.5% | 7.4% | 6.3% | 11.8% | 28.0% |

Thus, it is possible to conclude that almost 47% of the contribution to the model comes from the age, 28% from comorbidities and the remaining 25% from the other three.

**Table 5. Model built from all significant effects and with the training dataset.**

| Variable | Estimated Coeff | Standard Error | z | p-value | OR (95% CI) |
|---|---|---|---|---|---|
| (Intercept) | -1.673 | 0.019 | -86.69 | <0.001 | |
| age | 1.206 | 0.023 | 53.18 | <0.001 | 3.34 (3.19–3.49) |
| gender | 0.134 | 0.018 | 7.38 | <0.001 | 1.14 (1.1–1.19) |
| hcu_occupancy | 0.154 | 0.013 | 11.59 | <0.001 | 1.17 (1.14–1.2) |
| health care unit | 0.279 | 0.014 | 19.93 | <0.001 | 1.32 (1.29–1.36) |
| Clinical diagnoses | 0.802 | 0.016 | 51.60 | <0.001 | 2.23 (2.16–2.3) |
| age:gender | 0.074 | 0.022 | 3.44 | <0.001 | 1.08 (1.03–1.12) |
| age: Clinical diagnoses | -0.391 | 0.020 | -19.99 | <0.001 | 0.68 (0.65–0.7) |

## Model assessment

**ROC curve.** To assess general model performance, we can compute the ROC AUC. In addition, it's possible to establish a cut-off probability and predict death or survival if the model probability is higher or lower than this reference value, respectively. Out of multiple

## Fit metrics - sensitivity vs specificity
### optimal cut-off, using Train data

**Fig 4. Sensitivity and specificity, plotted as a function of the cut-off probability.** The optimal cut-off is 0.287 and a ROC AUC for Training of 0.811 was obtained.

possible approaches, we chose to define this value as the probability for which the sensitivity is equal to the specificity (Fig 4).

To avoid test leakage, the cut-off was calculated using training data. Computing the ROC AUC using the test partition, we obtained a value of 0.8. The value is similar to the one obtained using the training partition, indicating the robustness of the model. Let's now see how the model performs for each variable, using test data.

To better understand the adjustment of the predictions to the observed risk, we can compute a confidence interval for each prediction. In terms of the probability, the lower and upper bounds of the confidence intervals (CI) are 99%. Moreover, the lower and upper CI bounds in the following graphs are the average lower and upper limits of the individuals in each variable group (Figs 5–7).

With almost 47% of model contribution, age is the most important of all studied factors in predicting death by COVID-19.

In general, the model responds well to reality, even using the test database. Comorbidities are the 2nd most important variable. Instead of seeing the effect on the clinical diagnoses (sum of all comorbidities scores in an episode), a useful but cryptic variable, we present the results per comorbidity. We highlight septicemia as the most severe.

Is medical care better for hospitals with lower hcu occupancy? And if so, is it enough to have any influence on the mortality? Although not by much, as we can attest by the scale of variations in probability, we can in fact observe a risk increase associated with a higher hcu occupancy.

**Propensity of death - observed vs predicted**

by Age, using Test data

**Fig 5. Average predicted probability and observed death ratio for each age group, using test data.**

In summary, the model generally managed to capture the diversity of the data, and can be a useful tool to help determine, for example, when a patient should or not be admitted to the ICU (intensive care unit).

**Plotting multiple variables.** To visualize the optimal cut-off in the variables space, we built a graph of the sum of comorbidity scores vs age, the two most important variables in the model (Fig 8). Given that the model uses 5 variables to generate the probability of death the 3 remaining variables (gender, hcu occupancy score and hcu score) must be assigned to some value. In this case, we chose the 25th, 50th and 75th quantiles of hcu score and hcu occupancy' distributions. The 50th quantile values were used to build the main threshold line, while the 25th and 75th used for a pseudo confidence interval (favourable and unfavourable values of these two variables). Two threshold lines were built for each gender. Fig 8 shows the effect of gender in the threshold: the male boundary line is situated at younger ages compared to the female boundary; an indication of the higher risk present in males.

**Risk of death by comorbidity profile.** Another way to explore the effect of the comorbidities in the model is to create disease profiles (Fig 9). Thus, we establish 6 disease profiles and compute the corresponding sum of the comorbidity scores. Besides the disease profile, we can vary the age and gender in the analysis and use the mean hcu occupancy and mean healthcare unit score of the training partition. The profiles and predicted risk are the following:

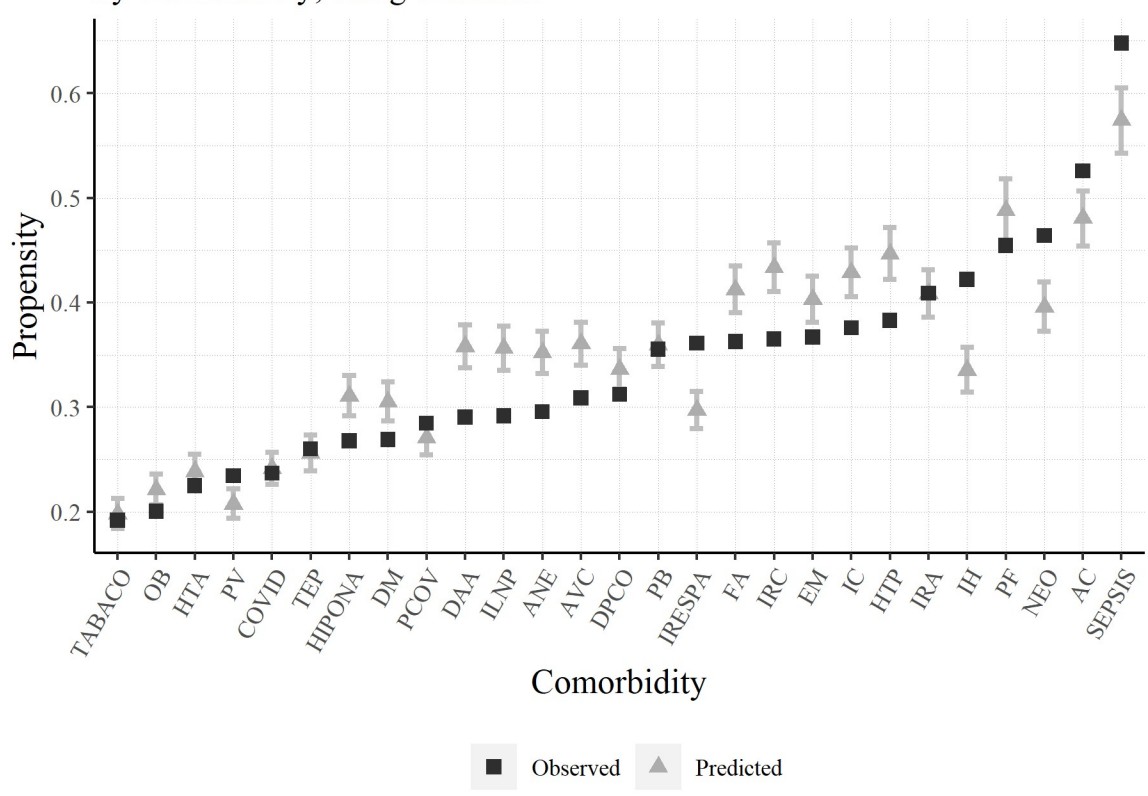

**Fig 6. Average predicted probability and observed death ratio for each comorbidity group, using test data.**

*Profile 1*: Just covid;

*Profile 2*: Covid and Chronic respiratory disease (DPCO);

*Profile 3*: Covid, diabetes mellitus (DM), arterial hypertension (HTA) and obesity (OB);

*Profile 4*: Covid, bacterial pneumonia (PB) and pneumonia to SARSCoV (PCOV);

*Profile 5*: Covid, pulmonary thromboembolism (TEP), arterial hypertension (HTA) and pneumonia to SARSCoV (PCOV);

*Profile 6*: Covid, acute breathing insufficiency (IRESPA), cardiac insufficiency (IC) and pneumonia to SARSCoV (PCOV).

In Fig 9, like in Fig 8, we can see a clear difference between the genders risk-wise. For example, at the age of 90, the worst female profile has a lower risk than the best male profile.

## Discussion and conclusions

This type of work is integrated within Risk Prediction models, a subject contained in hundreds of existing papers. In 2011, there were almost 8000 citations reviewed [13]. With this many studies involving risk prediction, what are our new contributions to the subject? The main contribution of this study relates to the quantity and originality of the data. Using a simple

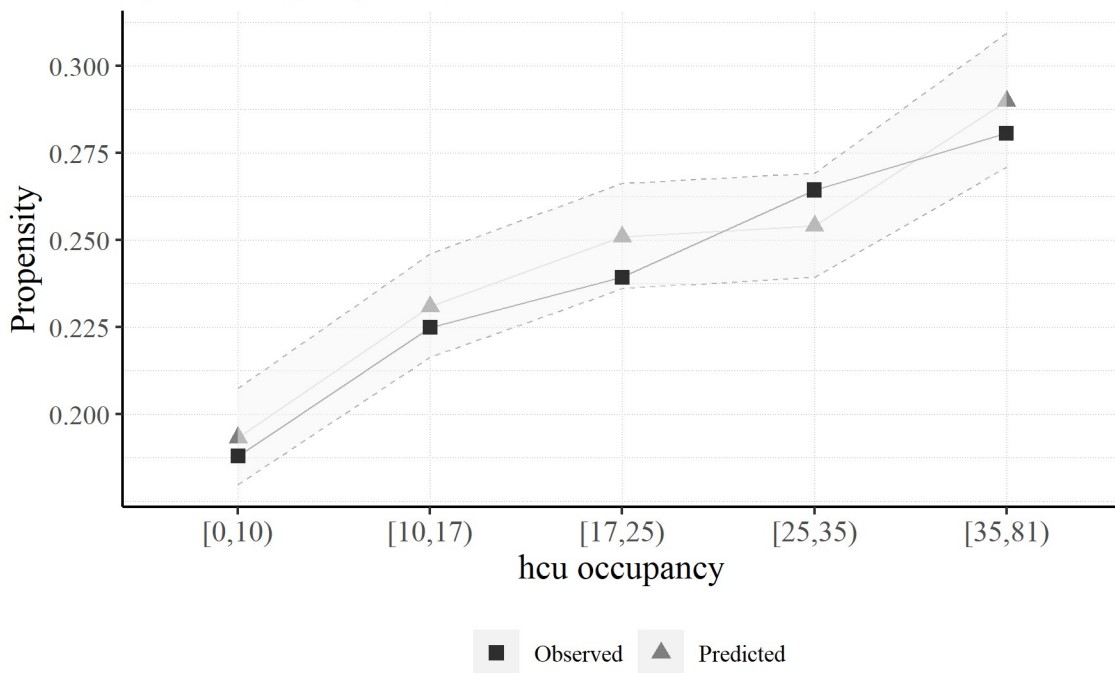

**Fig 7. Average predicted probability and observed death ratio for each hcu occupancy group, using test data.**

linear model, we managed to integrate the patient comorbidities, hcu occupancy and inter-hcu variability using score techniques, with commonly used variables (age, gender). This methodology allows us to weigh the relative importance of each variable in predicting death by COVID.

In this study we are looking at a time before vaccination was available, making it a useful tool to understand the efficacy of the vaccine. As soon as more recent data can be obtained, we will be able to analyse the results obtained for a vaccinated versus unvaccinated population.

Despite being a Portuguese database, this study is relevant to other countries (for instance, we expect the contributions of the 5 variables to the model to be similar using other countries' data, although we still recommend using, if possible, country specific data).

Given that age is almost 47% of importance in our model, it is interesting to consider whether Portugal (and other countries) should have applied their restrictive measures with more emphasis on age stratification, such as lockdowns (especially given their catastrophic economic and social effects [2, 3]).

The second most important variable (28%) is the sum of the comorbidities scores per episode. The analysis by patient profile also reveals the effect of the variable, with a substantial increase in the model probability for certain clinical profiles.

Several studies (for example [14]) indicate obesity as an extremely relevant factor. However, our Portuguese data do not allow us to draw this conclusion (in fact, in this database patients tagged with obesity have a lower death ratio than those who are not).

In terms of future work, it would be interesting to see if the model performance is preserved with data from May 2021 onwards. If not, we could then assess the causes, such as changes in

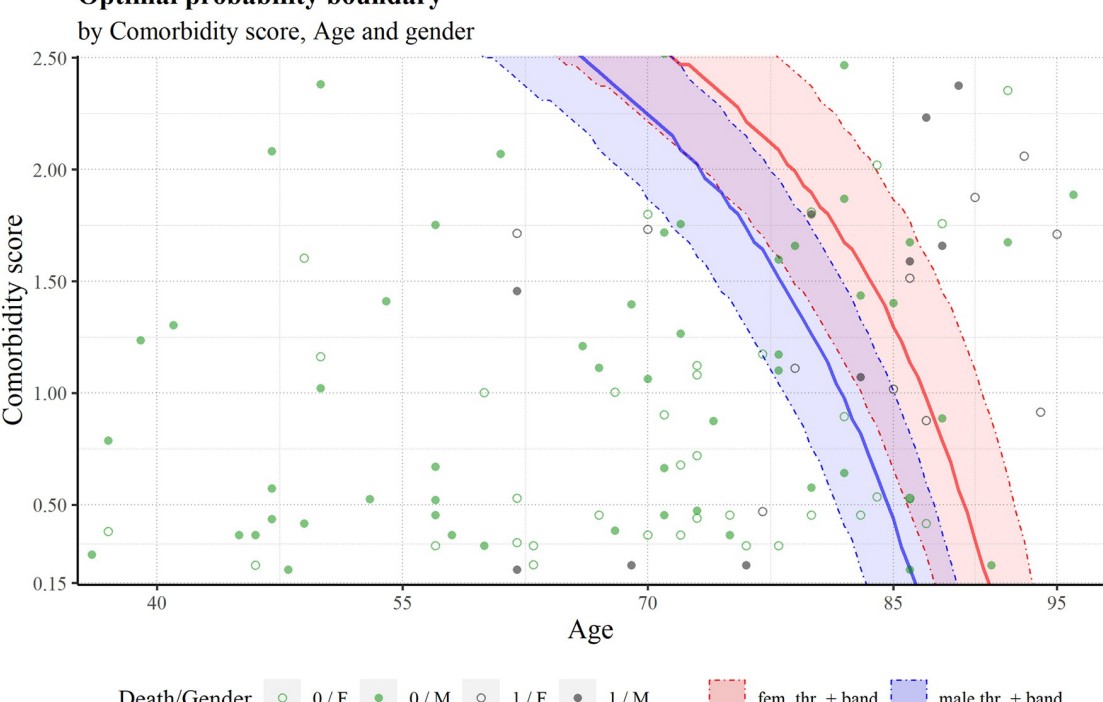

**Fig 8. Boundary using cut-off = 0.287 for female and male individuals, as a function of comorbidity score sum and age.** The left, center and right band limits correspond to the 25th, 50th and 75th quantile values of the hcu occupancy and hcu scores. Some test data is also plotted, along with the outcome, death (1) or survival (0).

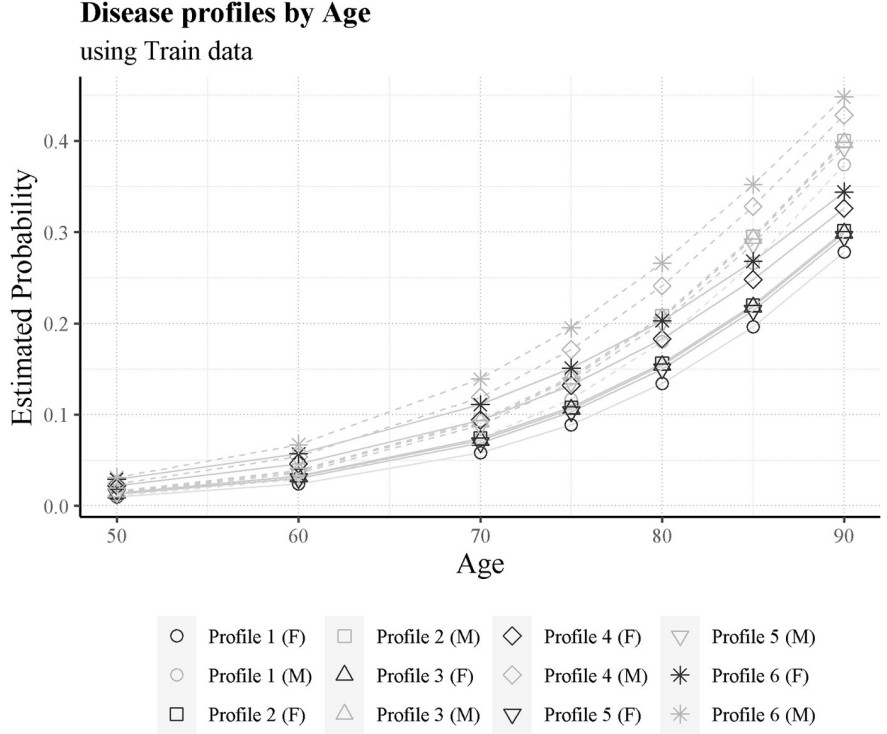

**Fig 9. Model probability for the 6 disease profiles, differentiated by age and gender.**

the mortality of the disease (due to new virus variants for instance), altering the pattern associated with age, comorbidity and possibly gender, or changes in hcu logistic, altering the effect of hcu occupancy and inter-hcu variability.

## Acknowledgments

The authors would like to thank

- Manuel do Carmo Gomes for providing the data without which this study would not have been possible.

- Sérgio Daniel das Neves Pereira Naito for his help and consulting on using a scorecard model approach for the construction of our model.

- John McDermott for his help with wording and English grammar. He received his Ph.D. in history at the University of Toronto and taught modern European history at the University of Winnipeg and McMaster University. He is now living in Portugal doing some editing when called upon.

## Author Contributions

**Conceptualization:** João José Ferreira Gomes.

**Formal analysis:** João José Ferreira Gomes, António Ferreira.

**Investigation:** João José Ferreira Gomes.

**Methodology:** João José Ferreira Gomes, Afonso Alves.

**Software:** Afonso Alves, Beatriz Nogueira Sequeira.

**Supervision:** João José Ferreira Gomes.

**Validation:** João José Ferreira Gomes, Afonso Alves.

**Visualization:** João José Ferreira Gomes, Afonso Alves.

**Writing – original draft:** João José Ferreira Gomes, Afonso Alves, Beatriz Nogueira Sequeira.

**Writing – review & editing:** João José Ferreira Gomes, Afonso Alves, Beatriz Nogueira Sequeira.

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
