## [Editor Report · Decision Letter 0]

11 Aug 2022

PONE-D-22-14646

A risk scoring model of COVID-19 at hospital admission

PLOS ONE

Dear Dr. Gomes ,

Thank you for submitting your manuscript to PLOS ONE. After careful consideration, we have decided that your manuscript does not meet our criteria for publication and must therefore be rejected.

Specifically:

1. The manuscript isn't organised scientifically, and the tables and figures isn't presented in academic style that are suitable for publication. 

2. Lack of details re the setting and conduction of study and poor interpretation of results. 

I am sorry that we cannot be more positive on this occasion, but hope that you appreciate the reasons for this decision.

Kind regards,

Jiaxu Zeng, Ph.D

Academic Editor

PLOS ONE

---

## [Author Response · Author response to Decision Letter 0]

24 Aug 2022

"1. The manuscript is not scientifically organized, and the tables and figures are not presented in an academic style that is suitable for publication."

The article is organized in the standard way for many scientific articles: Introduction, Methods, Results, Discussion and Conclusions.

Tables and figures were attached according to the guidelines suggested by PlosOne. The tables were incorporated in the text in the simplest possible way so that they could be adapted by the editor if necessary and the figures were attached in separate files as the submission manual suggests. In the new proposed manuscript, we altered 3 tables that were in the format of an R output image.

"2. Lack of details on the setting and conduction of study and poor interpretation of results."

This study was conducted from a database of over 50,000 people, of which over 12,000 individuals have died.

It contains the following information: Date of birth, date of entry and exit from the hospital, sex, health unit where the patient was admitted (which allowed us to, based on the information from the Portuguese Ministry of Health regarding hospital capacity, infer the pressure in terms of average hospital capacity per hospitalization and use this information in the model) and the various comorbidities associated with each patient (through a sophisticated process it was also possible to assign a risk score to each patient) – all this is described in detail in the manuscript.

Based on this material, we built a regression model that allows us to estimate the probability of death for each patient diagnosed with COVID and associate this probability to its respective confidence interval. In addition to that, we built profiles for the most frequent patterns with an associated probability of death (survival) to each, allowing us to have an idea of the increase in risk for a given factor. Furthermore, we retrospectively simulated some of these profiles to understand the model's ability to discriminate between survivals and deaths. It was also possible to objectively quantify the importance of age in terms of COVID mortality - about 47% compared to the 5 factors available.

We hope to have been able to enlighten the importance of this study and the results obtained, specially for an in-depth analysis of the strategy that was followed by most countries during the pandemic outburst.

---

## [Decision Letter · Decision Letter 1]

22 Sep 2022

PONE-D-22-14646R1

A risk scoring model of COVID-19 at hospital admission

PLOS ONE

Dear Dr. Gomes,

Thank you for submitting your manuscript to PLOS ONE. After careful consideration, we feel that it has merit but does not fully meet PLOS ONE’s publication criteria as it currently stands. Therefore, we invite you to submit a revised version of the manuscript that addresses the points raised during the review process.

Please submit your revised manuscript by 11/15/2022. If you will need more time than this to complete your revisions, please reply to this message or contact the journal office at plosone@plos.org. Please include the following items when submitting your revised manuscript:

We look forward to receiving your revised manuscript

Kind regards,

Yuyan Wang, Ph.D.

Academic Editor

PLOS ONE

Journal Requirements:

3. Please note that PLOS ONE has specific guidelines on code sharing for submissions in which author-generated code underpins the findings in the manuscript. In these cases, all author-generated code must be made available without restrictions upon publication of the work. Please review our guidelines at https://journals.plos.org/plosone/s/materials-and-software-sharing#loc-sharing-code and ensure that your code is shared in a way that follows best practice and facilitates reproducibility and reuse

4.Your abstract cannot contain citations. Please remove any existing citations from the abstract section. You may only include citations in the body text of the manuscript, and please ensure that they remain in ascending numerical order on first mention.

Additional Editor Comments (if provided):

Please refer to some published papers, revise the manuscript in an academic way and make it more rigorous.

Please note that Review 2 was provided by the Academic Editor, Yuyan Wang

Reviewers' comments:

Reviewer's Responses to Questions

**Comments to the Author**

1. If the authors have adequately addressed your comments raised in a previous round of review and you feel that this manuscript is now acceptable for publication, you may indicate that here to bypass the “Comments to the Author” section, enter your conflict of interest statement in the “Confidential to Editor” section, and submit your "Accept" recommendation.

Reviewer #1: All comments have been addressed

Reviewer #2: (No Response)

2. Is the manuscript technically sound, and do the data support the conclusions?

Reviewer #1: Yes

Reviewer #2: Yes

3. Has the statistical analysis been performed appropriately and rigorously?

Reviewer #1: Yes

Reviewer #2: No

4. Have the authors made all data underlying the findings in their manuscript fully available?

Reviewer #1: Yes

Reviewer #2: Yes

5. Is the manuscript presented in an intelligible fashion and written in standard English?

Reviewer #1: Yes

Reviewer #2: No

6. Review Comments to the Author

Reviewer #1: This is an interesting well written manuscript on an important topic.

In my opinion al comments have been answered so the paper can be published

Reviewer #2: The authors used logistic regression to predict the death probability for inpatients diagnosed with COVID. The topic is interesting, however, the manuscript is not easy to read and the authors should refer to published articles to make the paper more rigorous not only in content but also in the presented results.

Major:

1. I would suggest the authors look for some professional English editorial help to avoid grammar errors and make the manuscript fluent and easier to read.

2. Abstract: the Conclusion part is missing in Abstract.

3. The way the authors created the diagnoses score is unclear. More details (or specific example how the scores are creasted) are needed.

4. The context in the manuscript is too messy. Please refer to the published paper (from PLOS ONE or any other academic journals) to clean up the Methods and Results section. Specifically,

a. Row 76 to 79: it will be good to have a data flowchart to reflect the subject number change by inclusion or exclusion creation.

b. Row 90-99: it’s unnecessary to show the variable name from original dataset

c. Row 101: Table 1 are unnecessary and can be moved to supplementary tables if the authors insist to keep them.

d. Row 108: Table 2, there are too many facilities and it would better to use a bar plot to show the number of patients and death rate.

e. Table 3, 4 and 5 can be combined as one descriptive table and should be placed in Results section.

f. Row 125: Table 6 can be merged into Table 2 and presented as plot as well.

g. Row 152: Table 7 isn’t presented in academic style. ORs wit CIs are better.

h. Why the presented results content are different in Table 8 (Df/Deviance/Resid.Df…) and Table 9 (Estimate/Std./Error)? These results are not presented in standard way. No one should directly present formula and output results from analytical software. Please refer to the published paper to make it more rigorous.

i. Row 169: could you cite reference to use “percentage by dividing each coefficient by the sum of all coefficients” to describe the variable importance?

j. Row 179: Table 11 needs to be revised like Table 9, and the interpretation needs improvement.

k. Row 223: Figure 5 and Figure 6 can be merged as one plot to show the difference of the boundary line between male and female.

l. Row 242: Table 12 can be transformed as a plot using Age as x-axis and probability as y-axis. Then it will be easier to compare among different profile and gender groups.

7. PLOS authors have the option to publish the peer review history of their article (what does this mean?). If published, this will include your full peer review and any attached files.

Reviewer #1: No

Reviewer #2: No

---

## [Author Response · Author response to Decision Letter 1]

25 Nov 2022

Review Comments

1. “I would suggest the authors look for some professional English editorial help to avoid grammar errors and make the manuscript fluent and easier to read”

The writing has been carefully revised as suggested by reviewer.

2. “Abstract: the Conclusion part is missing in Abstract”

Conclusions were added to the Abstract section of the paper.

3. “The way the authors created the diagnoses score is unclear. More details (or specific example how the scores are created) are needed.”

The approach applied followed a scorecard model approach, which is meant to be simpler to explain and apply. Two of the most common cited references in the literature are, for instance: [Thomas, Edelman, Crook; Credit Scoring and Its Applications Society for Industrial and Applied Mathematics (1983)] [Anderson, R.; Credit Scoring Toolkit - Theory and Practice for Retail Credit Risk Management and Decision Automation, Oxford University Press (2007)].

Further explanation on this topic was added to the manuscript in the Methods section.

4. “The context in the manuscript is too messy. Please refer to the published paper (from PLOS ONE or any other academic journals) to clean up the Methods and Results section. Specifically:”

a) “Row 76 to 79: it will be good to have a data flowchart to reflect the subject number change by inclusion or exclusion creation”

A flowchart was added to the manuscript, explaining the transformations made to the initial database.

b) “Row 90-99: it’s unnecessary to show the variable name from original dataset”

The names presented correspond to the variable names used in our model and a brief description clarifying what each variable means. However, presentation has been simplified.

c) “Row 101: Table 1 are unnecessary and can be moved to supplementary tables if the authors insist to keep them.”

Table 1 was removed according to suggestion.

d) “Row 108: Table 2, there are too many facilities and it would better to use a bar plot to show the number of patients and death rate.”

Information previously presented in Table 2 was converted to a bar plot (Fig. 2), per suggestion.

e) “Table 3, 4 and 5 can be combined as one descriptive table and should be placed in Results section.”

The suggested changes were made – Tables 3,4 and 5 were combined into one single table (Table 1) and were moved to the Results section.

f) “Row 125: Table 6 can be merged into Table 2 and presented as plot as well”

Tables 2 and 6 have been combined as suggested as were converted into a bar plot (Fig. 3).

g) “Row 152: Table 7 isn’t presented in academic style. ORs with CIs are better”

h) “Why the presented results content are different in Table 8 (Df/Deviance/Resid.Df…) and Table 9 (Estimate/Std./Error)? These results are not presented in standard way. No one should directly present formula and output results from analytical software. Please refer to the published paper to make it more rigorous.”

Tables 7 and 8 have been combined (Table 2) and the results of the univariate analysis are now presented in a standard way, per suggestion.

i) “Row 169: could you cite reference to use “percentage by dividing each coefficient by the sum of all coefficients” to describe the variable importance?”

Further explanation as to why this approach was followed was added to the manuscript, as well as reference (Multiple Regression section in Results), as suggested.

j) “Row 179: Table 11 needs to be revised like Table 9, and the interpretation needs improvement.”

Tables 9 and 11 were revised and the results are now presented in a standard way (they now correspond to Tables 3 and 5, respectively). Interpretation of the results has also been improved.

k) “ Row 223: Figure 5 and Figure 6 can be merged as one plot to show the difference of the boundary line between male and female.”

Figures 5 and 6 have been merged into one single plot (Fig. 8)

l) “Row 242: Table 12 can be transformed as a plot using Age as x-axis and probability as y-axis. Then it will be easier to compare among different profile and gender groups.”

Table 12 has been transformed into a plot, per suggestion.

---

## [Decision Letter · Decision Letter 2]

8 Mar 2023

PONE-D-22-14646R2A risk scoring model of COVID-19 at hospital admissionPLOS ONE

Dear Dr. Gomes,

Thank you for submitting your manuscript to PLOS ONE. After careful consideration, we feel that it has merit but does not fully meet PLOS ONE’s publication criteria as it currently stands. Therefore, we invite you to submit a revised version of the manuscript that addresses the points raised during the review process.

Please revise.

We look forward to receiving your revised manuscript.

Kind regards,

Academic Editor

PLOS ONE

Reviewers' comments:

Reviewer's Responses to Questions

**Comments to the Author**

1. If the authors have adequately addressed your comments raised in a previous round of review and you feel that this manuscript is now acceptable for publication, you may indicate that here to bypass the “Comments to the Author” section, enter your conflict of interest statement in the “Confidential to Editor” section, and submit your "Accept" recommendation.

Reviewer #3: (No Response)

Reviewer #4: (No Response)

2. Is the manuscript technically sound, and do the data support the conclusions?

Reviewer #3: Partly

Reviewer #4: Yes

3. Has the statistical analysis been performed appropriately and rigorously? 

Reviewer #3: No

Reviewer #4: Yes

4. Have the authors made all data underlying the findings in their manuscript fully available?

Reviewer #3: Yes

Reviewer #4: Yes

5. Is the manuscript presented in an intelligible fashion and written in standard English?

Reviewer #3: No

Reviewer #4: Yes

6. Review Comments to the Author

Reviewer #3: A risk scoring model of COVID-19 at hospital admission. This paper analyzed how patient baseline characteristics and healthcare units’ characteristics could impact the mortality of covid in Portugal. A scoring method was utilized before the logistic regression.

Overall, this paper needs to be better written in English. I could only understand ~80% of this paper. Hence, the authors should seek professional help in the wording, grammar, and correcting the typos.

The paper models the probability of survival on both patient baseline characteristics/ and the characteristics of healthcare institutes. A score transformation was used on the predictive variables. However, is it necessary? Logistic regression can model both categorical and continuous predictive variables. And with an l-1 penalty, it can further handle variable selection. Authors should explain the motivation and benefit of using this scoring pre-process method.

Method section where variables were transformed into continuous scores, please add some explanations in the context. Despite citing two articles, I believe these methods are less well-known than the other statistical methods. Thus more details could help readers understand and even replicate this research.

Method: the hcu beds were excluded; however, such exclusion was not well justified.

Method: was modeled as categorical? In table 1, the author chose ten levels for age. Suggest keeping the level of categorical beyond 4. Otherwise, it might be more beneficial to model it as a continuous variable directly. If it was modeled as continuous, suggest reporting the mean/sd and Median(Range) for this variable in Table 1.

Data- 52,400 patients but 52390 episodes? What’s the difference between the ID and patients?

Data - hcu pressure(average hcu occupancy rate); what is the definition or math formula for hcu occupancy rate? Is it derived from stay duration and the number of beds?

P5-R13: please define the weights-of-evidence transformation (WOE) method. It would be helpful to add more details.

Table 1- only three variables were reported; what about the HCU-related two variables? It would be more straightforward to report the variable by event, i.e., two columns in the table as death or survival; cell value could be either mean/sd + median/range or percentage.

Figure 1- add numbers being excluded from each step.

Figure 3- Title mismatch legends. Is the blue bar number of occupation rate or healthcare unit pressure? Please reconcile; only one name is needed for one variable.

Minor comments:

1. the manuscript needs to be better formatted; for example, rows 81, 85, 102, 144, and 271 are in different fonts from the rest of the paper. Please reconcile it.

3. Variables were bolded in font but not needed.

4. Variable names: hcu pressure could be replaced by a better/clear variable name.

5. It would be interesting to see how the covariates would predict the time from admission to death.

Reviewer #4: This article addresses an important public health issue which continue to impact healthcare around the world, despite > 3 yaers of experience in dealing with COVID 19 pandemic. The pre-vaccination period of this study shows the factors leading to death at a pospulation level. It address the unique feature of public sector healthcare units and impact of their size & bed occpancy on a pandemic of this magnitude. It also gives important data for public health authorities in terms of prepardness for the future pandemics. There are some imporatnt short comings in the manuscript that need addressing.

1, Overall size of the manuscript is too long and the proportion of different sections isn't uniform. e.g. Discussion section is relatively small as compared methodology which is very detailed and "wordy" I would recommend to cut down the methodology section.

2. The information within each section isn't appropriate for the respective section. e.g. sentence 68-70 should be in methodology. point 72-74 should be in discussion section. point 95-96 should be in results section

3. exclusion of so called "reargard hospital" oncology units and patients receiving hemodialysis would have lead to exclusion of an important vulnerable group with high expected mortality.

7. PLOS authors have the option to publish the peer review history of their article (what does this mean?). If published, this will include your full peer review and any attached files.

Reviewer #3: No

Reviewer #4: **Yes: **msalbur

---

## [Author Response · Author response to Decision Letter 2]

18 Apr 2023

Please, see the enclosed response for the reviewers

---

## [Decision Letter · Decision Letter 3]

29 Jun 2023

A risk scoring model of COVID-19 at hospital admission

PONE-D-22-14646R3

Dear Dr. Gomes,

We’re pleased to inform you that your manuscript has been judged scientifically suitable for publication and will be formally accepted for publication once it meets all outstanding technical requirements.

Kind regards,

Academic Editor

PLOS ONE

Additional Editor Comments (optional):

Reviewers' comments:

Reviewer's Responses to Questions

**Comments to the Author**

1. If the authors have adequately addressed your comments raised in a previous round of review and you feel that this manuscript is now acceptable for publication, you may indicate that here to bypass the “Comments to the Author” section, enter your conflict of interest statement in the “Confidential to Editor” section, and submit your "Accept" recommendation.

Reviewer #4: All comments have been addressed

Reviewer #5: All comments have been addressed

2. Is the manuscript technically sound, and do the data support the conclusions?

Reviewer #4: Yes

Reviewer #5: Yes

3. Has the statistical analysis been performed appropriately and rigorously? 

Reviewer #4: Yes

Reviewer #5: Yes

4. Have the authors made all data underlying the findings in their manuscript fully available?

Reviewer #4: Yes

Reviewer #5: Yes

5. Is the manuscript presented in an intelligible fashion and written in standard English?

Reviewer #4: Yes

Reviewer #5: Yes

6. Review Comments to the Author

Reviewer #4: I am very pleased to see that you have made all the suggestions / recommendations from me. Also you have addressed some of the general reconfiguration of the manuscript.

Reviewer #5: Manuscript is completely revised and ready to get published. All the subjects in this article is addressed appropriately and needs no more revision.

7. PLOS authors have the option to publish the peer review history of their article (what does this mean?). If published, this will include your full peer review and any attached files.

Reviewer #4: **Yes: **M S Albur

Reviewer #5: No

---

## [Editor Report · Acceptance letter]

12 Jul 2023

PONE-D-22-14646R3 

A risk scoring model of COVID-19 at hospital admission 

Dear Dr. Gomes:

I'm pleased to inform you that your manuscript has been deemed suitable for publication in PLOS ONE. Congratulations! Your manuscript is now with our production department. 

Kind regards, 

on behalf of

Dr. Robert Jeenchen Chen 

Academic Editor

PLOS ONE